# Non-lithifying microbial ecosystem dissolves peritidal lime sand

Theodore M. Present [1 ✉], Maya L. Gomes [2], Elizabeth J. Trower [3], Nathan T. Stein[1], Usha F. Lingappa[1], John Naviaux [1], Michael T. Thorpe[4], Marjorie D. Cantine [5], Woodward W. Fischer[1], Andrew H. Knoll[6] & John P. Grotzinger[1]

Microbialites accrete where environmental conditions and microbial metabolisms promote lithification, commonly through carbonate cementation. On Little Ambergris Cay, Turks and Caicos Islands, microbial mats occur widely in peritidal environments above ooid sand but do not become lithified or preserved. Sediment cores and porewater geochemistry indicated that aerobic respiration and sulfide oxidation inhibit lithification and dissolve calcium carbonate sand despite widespread aragonite precipitation from platform surface waters. Here, we report that in tidally pumped environments, microbial metabolisms can negate the effects of taphonomically-favorable seawater chemistry on carbonate mineral saturation and microbialite development.

[1] Division of Geological and Planetary Sciences, California Institute of Technology, Pasadena, CA, USA. [2] Department of Earth and Planetary Sciences, Johns Hopkins University, Baltimore, MD, USA. [3] Department of Geological Sciences, University of Colorado Boulder, Boulder, CO, USA. [4] NASA Postdoctoral Program, NASA Johnson Space Center, Houston, TX, USA. [5] Department of Earth, Atmospheric, and Planetary Sciences, Massachusetts Institute of Technology, Cambridge, MA, USA. [6] Department of Organismic and Evolutionary Biology, Harvard University, Cambridge, MA, USA. ✉email: ted@caltech.edu

Microbialites—sedimentary rocks with fabric that reflects the influence of benthic microbial communities[1]—capture biogeochemical interactions between microbes and their environment throughout Earth's history, e.g. refs. [2,3]. Microbial mats are often considered steady-state ecosystems whose constituents consume each other's metabolic products[4]. To be preserved as microbialites, organic trapping and binding of sediment alone are insufficient because eventually, organisms in the lower parts of the mat consume the material binding grains[5]. Canonically, then, propitious microbial metabolisms drive microbialite formation by promoting authigenic cementation, e.g. refs. [6–8]. Lithification occurs most readily where microbial processes within the mat locally increase carbonate mineral saturation, typically in net productive environments where photosynthesis exceeds respiration and fermentation rates[9–11].

Benthic, peritidal, microbial mats grow widely on Little Ambergris Cay, Turks and Caicos Islands, which formed in the lee of Big Ambergris Cay at the eastern end of a 20 km-long ooid shoal, near the eastern margin of the Caicos platform (Fig. 1a)[12–15]. The shoal is characterized by well-sorted, medium sand-sized ooids with hardgrounds, grapestone sand, and cemented intraclasts on its northern flank[12,16–18]. Little Ambergris Cay (Fig. 1b) grew through shoreface accretion of ooid sand[19]. Holocene eolian grainstone ridges rim a shallow interior basin that contains sand flats with widespread microbial mats (Fig. 1c)[12,19–23]. Foreshore and interior basin sands are less well-sorted and contain more skeletal grains than the shoal, especially where concentrated in tidal lags[16,17,19].

Three microbial mat types occur in the interior basin: smooth mats, tufted polygonal mats, and colloform blister mats[19,20]. In the shallowest subtidal parts of the interior basin, smooth mats form pigmented mucilaginous blankets covered in beige extracellular polymeric substances (Fig. 1d). On intertidal sand flats and around mangrove thickets, polygonal mats form dark green, decimeter-scale, domal biscuits, often with millimeter-scale, dark

gray-green, filamentous tufts (Fig. 1e). In supratidal environments on interior flanks of the eolian ridges, blister mats form centimeter-scale, fenestral, dark green to gray, colloform bumps (Fig. 1f). Mats cover more than 25% of the area on Little Ambergris Cay and more than 40% of the interior basin, and they have been observed to rapidly regrow on disturbed surfaces after storm scouring[19]. Discontinuous, cemented ooid crusts less than 1 cm thick underlie some mats, but many mats grow on poorly cohesive ooid sand. The upper few millimeters of ooid sands are often stabilized by microbial communities that may desiccate to form the crusts or be dislodged to form flat, angular intraclasts. All three mat types contain similar microbial communities, indicating that their textures reflect environmental controls more so than taxonomy[19–22]. The mats contain benthic foraminifera and cerithid gastropods (Fig. 1f), and calcareous (e.g., *Halimeda* and *Penicillus*) and non-calcareous algae (e.g., *Bataphora*) live in subtidal portions of the interior basin.

Here, we show that the net metabolic activity of widespread microbial mats on Little Ambergris Cay inhibits—rather than promotes—lithification. Microbial mats, including those on Little Ambergris Cay, often grow in environments otherwise characterized by widespread carbonate precipitation, so why do they rarely form microbialites? Microbial metabolisms that inhibit lithification can only do so by consuming the products of other metabolisms that promote lithification—at worst, there can be no net diminishment of favorable ambient conditions for lithification. We find insight into this apparent paradox with sedimentological and geochemical evidence of widespread dissolution of calcium carbonate sand below the microbial mats.

## Results

**Sediments beneath microbial mats**. Eight vibracores were collected across Little Ambergris Cay in August 2017 (Fig. 1b and Supplementary Table 1). Vibracores, up to 3 m long, included

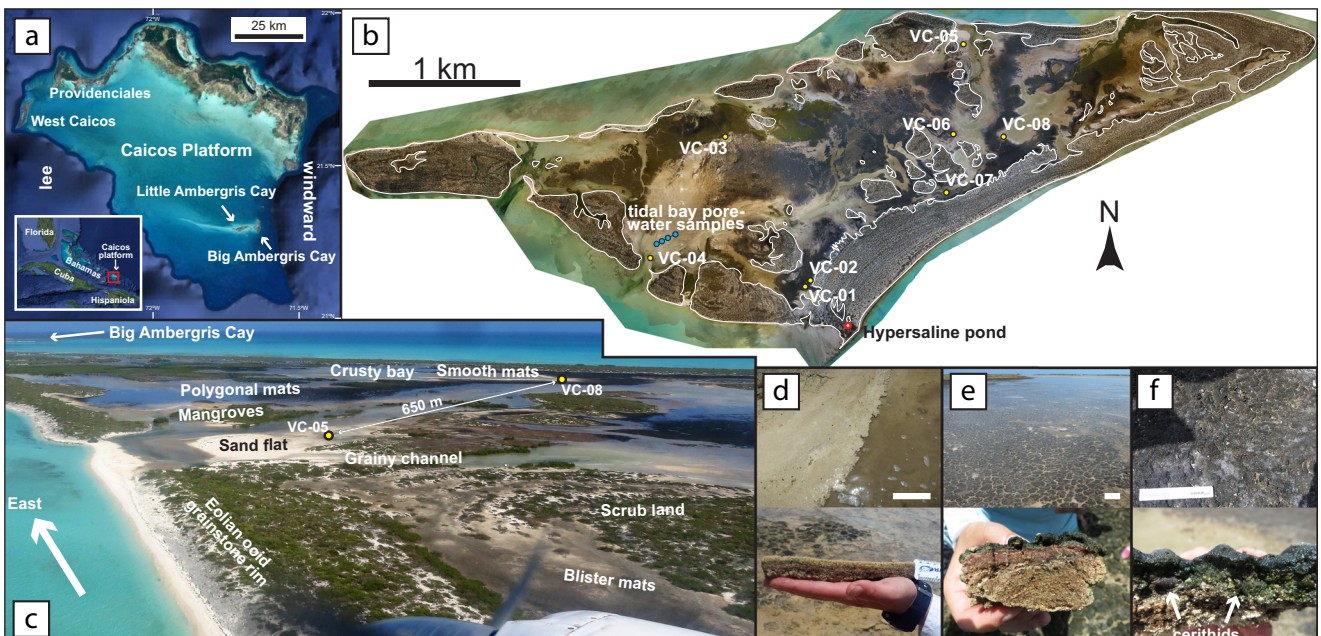

**Fig. 1 Microbial mats and their environment on Little Ambergris Cay. a** Caicos platform and its location in the southeastern Bahamian archipelago (inset). Satellite imagery from Google Earth[40]. **b** Little Ambergris Cay drone orthomosaic[19] marked with core, pond, and tidal bay porewater sampling locations. White contours mark the maximum tide level derived from tide monitoring stations and a photogrammetric digital elevation model[19], simplified to remove local bias from mangrove canopies above intertidal surfaces. **c** Aerial view southeast over interior basin rimmed by eolian grainstone ridge at the eastern end of Little Ambergris Cay. **d** Shallowest subtidal smooth mats on the edge of a tidal creek. **e** Intertidal polygonal mats. **f** Supratidal and upper intertidal blister mats with cerithid gastropods. Scales in top panels of **d**–**f** are 15 cm.

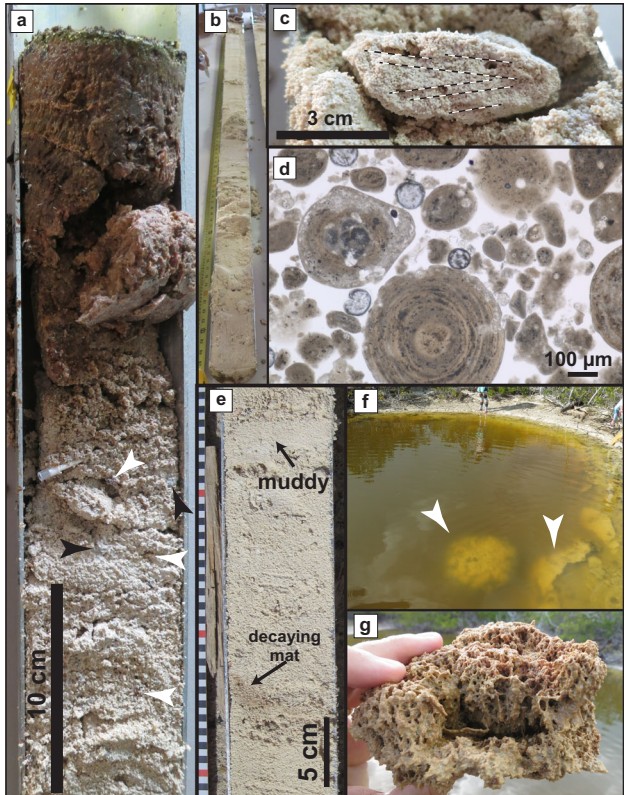

**Fig. 2 Sediments associated with microbial mats and crusts. a** Top of core VC-03 with polygonal microbial mat overlying ooid-skeletal sand with cerithid gastropods (black arrows) and bivalve fragments (white arrows). **b** VC-03 consists of ooid-skeletal sand throughout its 223 cm length. **c** Nodule in a partially indurated interval of core VC-03 at 178 cm preserving small burrows and ripple cross-lamination. **d** Plane-polarized light photomicrograph showing dissolution features in ooid-*Halimeda* sand in VC-03 at 150 cm. **e** Laminated organic material interpreted as a decaying mat in VC-05 at 60 cm, and a thin, mud-rich interval at 42 cm. **f** Calcifying mats (arrows) along the rim of the hypersaline pond. Researchers for scale. **g** Mineralization of mangrove leaves and filaments in the calcified mats rimming the pond.

active microbial mats up to 12 cm thick (Fig. 2a) above ooid sand with lesser skeletal sand to gravel and ooid grainstone intraclasts (Fig. 2b). Skeletal material was predominately disarticulated and/ or fragmented coralline red algae, calcareous green algae (mainly *Halimeda*), cerithid gastropods, venerid, and arcoid bivalves, benthic foraminifera, and serpulid worm tubes. Ooid grainstone intraclasts included angular, flat, 1 to 4 cm flakes and rounded, 3 to 5 cm nodules preserving cm-scale burrows and macroalgal holdfasts. Muddy ooid or foraminiferal sand layers represented less than 5% of the sediment and were 1 to 5 cm thick. Below about 1 m, the cores were partially indurated, occasionally preserving burrows, holdfasts, and ripple cross lamination (Fig. 2c). In thin sections, ooid-skeletal sand was heavily microbored and displayed pervasive fabrics indicative of dissolution, including solution-enlarged microbores, irregular grain boundaries, and selective leaching of micrite envelopes and micritized borings (Fig. 2d). Calcium carbonate $\delta^{13}C$ values in the sediment increased with depth from 4.9‰ to 5.2‰ in a supratidal core, from 3.8‰ to 5.1‰ in an intertidal core, and from 4.6‰ to 5.2‰ in a subtidal core (Fig. 3).

Apart from the active mats at core tops, visible organic material was mainly limited to rare plant debris. Beneath blister mats on the edge of a tidal channel (core VC-05), we found a single occurrence of faintly laminated organic material in a 2 cm-thick, tan, muddy, ooid-cerithid gastropod layer at 60 cm depth (Fig. 2e). This material displayed characteristic spectral features of chlorophyll *a* and bacteriochlorophyll *a* pigments—two main light-gathering pigments seen in the polygonal mats[19].

An isolated occurrence of calcified, decimeter-scale, bulbous domes was found rimming a 1.4 m-deep pond in the accretionary beach ridges on the windward-facing southern part of Little Ambergris Cay (Fig. 2f). The pond, surrounded by shrubland vegetation, was filled with tannic, salinity-stratified (43 to 48 ppt), hypersaline water. The domes comprised centimeter-thick fenestral palisades of carbonate-coated filaments and mangrove leaves (Fig. 2g). These mineralized bulbous domes were unlike any of the three mat textures elsewhere on the island. Decaying microbial mats along the pond floor near its rim were observed in summer 2016 where the crusts were found but were not present when the crusts were resampled in 2017; nearby ponds did contain active mats.

**Porewater geochemistry.** Porewater was sampled from 3 vibra-cores situated in low supratidal polygonal mats (VC-01), inter-tidal polygonal mats (VC-03), and a subtidal, muddy, *Batophora* algal substrate (VC-04) (Fig. 3) (Supplementary Data 1). Pore-waters ranged from normal marine to hypersaline (34 to 50 ppt). Salinities were invariant with depth and the highest evaporation factors relative to platform seawater (calculated from chlorinity data) were observed in the supratidal core, VC-01. Porewater dissolved inorganic carbon (DIC) concentrations, which did not change systematically with depth through the entirety of the supratidal blister mat core or the uppermost 1 m of the other cores, were all higher than seawater: approximately 2.5 mmol/kg below the blister mats, 5.9 mmol/kg below the polygonal mats, and 7.2 mmol/kg below the subtidal algae. Below 1 m, DIC increased to 10.4 mmol/kg in the polygonal mats and decreased to 4.2 mmol/kg in the subtidal algae. In all cores, alkalinity closely followed DIC. Resultant calculated aragonite saturation states ($\Omega$) also did not vary systematically with depth, were less than 2.5 in intertidal and subtidal cores, and exceeded 6 in the supratidal core. DIC $\delta^{13}C$ values increased with depth in each core: from −2.0‰ to −0.1‰ below the blister mats, from −5.4‰ to −1.2‰ below the polygonal mats, and from −3.2‰ to 0.1‰ below the subtidal algae. Chloride-normalized sulfate concentrations ranged from 17% lower to 18% higher than seawater, and total sulfide concentrations ranged from detection limits to 1.76 mM.

Stations for tidal bay porewater sampling at low and high tide were installed across a tidal bar and channel (Fig. 1b) at 0.2, 0.5, and 1 m depths (Supplementary Data 1). Tidal bay porewater was salinity-stratified at high tide, with *ca.* 40 ppt water at 1 m below the surface (Fig. 4). At low tide, tidal bay porewater was temperature-stratified, as surface waters warmed above 33 °C while water at 1 m depth only warmed to 29 °C. Tidal bay porewater chemistry varied substantially through a tidal cycle, with transient aragonite saturation indices as low as 0.1 and as high as 17 and sulfate depleted relative to chlorinity-normalized concentrations by up to −3.3%.

Water was also sampled at 7 depths from the hypersaline pond (43 to 48 ppt) at the southern end of the cay that contained the calcified domes (Fig. 1b) (Supplementary Data 1). The pond contained largely invariant DIC and alkalinity concentrations near *ca.* 6 mmol/kg (Supplementary Fig. 1). In the pond, small alkalinity fluctuations caused $\Omega$ to generally decrease with depth from 18.8 to 2.2. Sulfate was in excess of chlorinity-normalized concentrations by 2.8% to 6.2%.

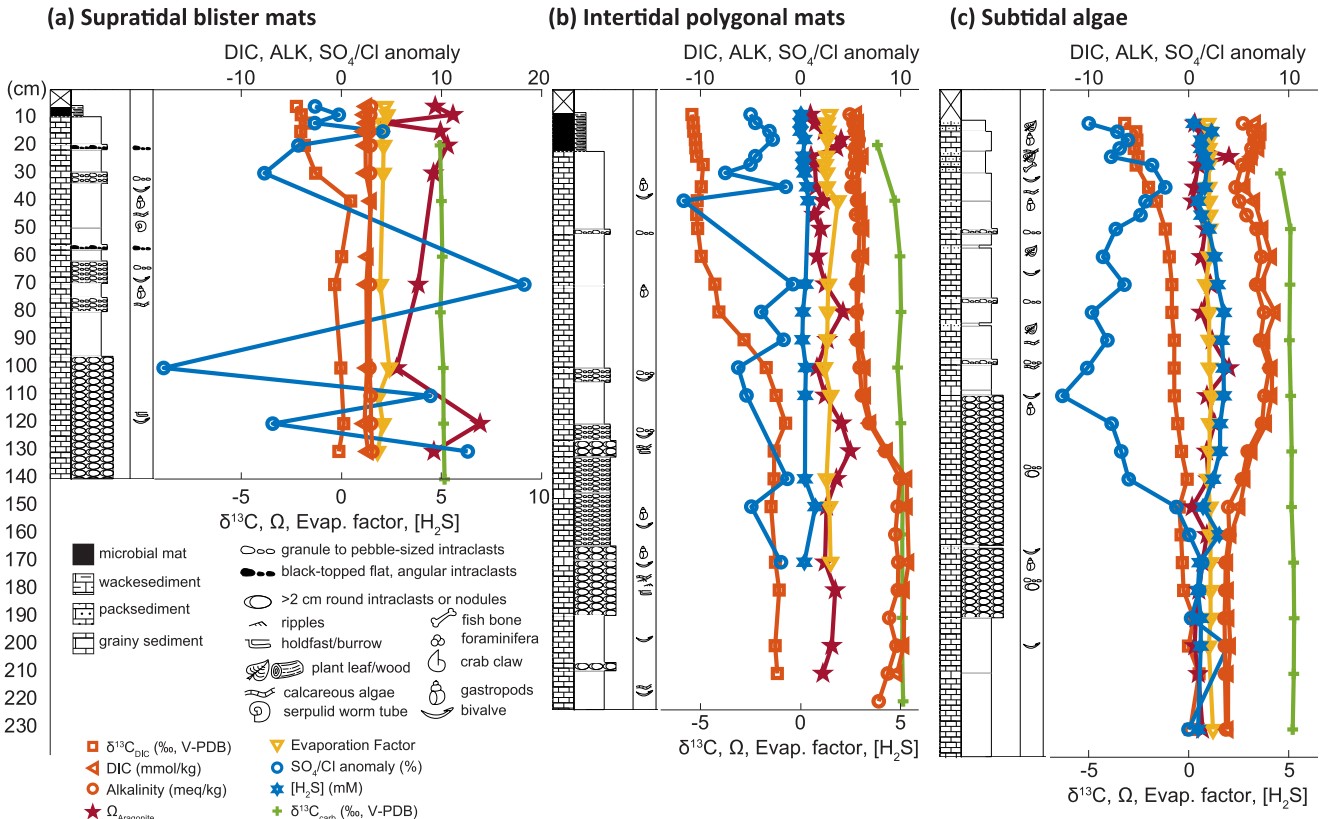

**Fig. 3 Sedimentology and geochemistry of vibracores.** Geochemical data and graphic logs show the effects of tidal pumping and biogeochemical processes on aragonite saturation. Additional logs from cores without porewater sampling are shown in Supplementary Fig. 2 and Supplementary Fig. 3. Evap. = evaporation, ALK = alkalinity, V-PDB = Vienna Pee Dee Belemnite. **a** Core VC-01, situated in low supratidal mats on the flank of the eolian grainstone berm. **b** Core VC-03, located in intertidal polygonal mats near a mangrove thicket. **c** Core VC-04, situated in a subtidal, muddy *Batophora* algae patch.

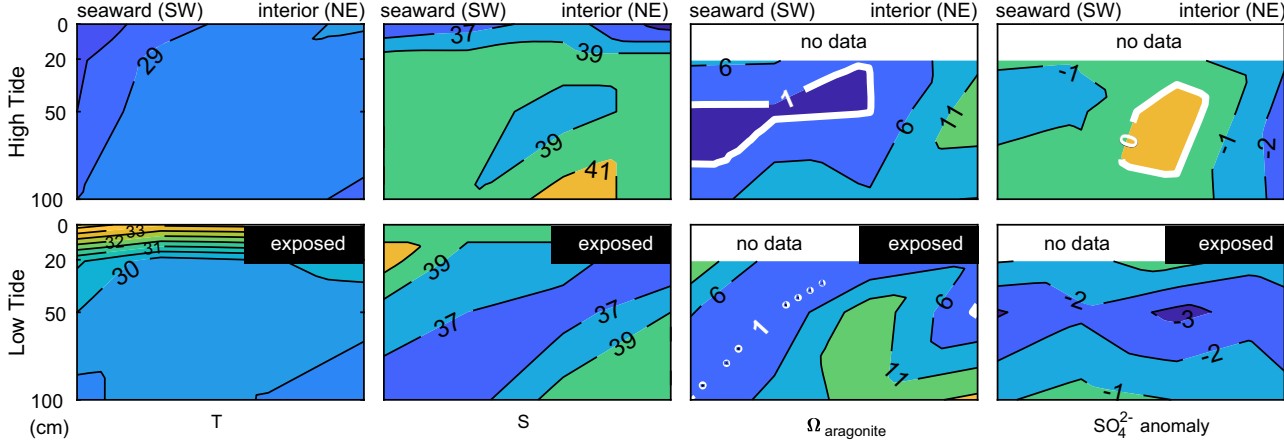

**Fig. 4 Tidal bay porewater geochemistry.** Contours show the low and high tide structure of tidal bay porewater temperature (T), salinity (S), aragonite saturation ($\Omega$), and sulfate anomaly at the interior basin inlet on the southwestern side of Little Ambergris Cay.

## Discussion

The Caicos platform exhibits widespread inorganic precipitation as ooids and hardgrounds and has surface seawater supersaturated with respect to aragonite[12,16,18]. These conditions should permit cementation that would preserve microbial mat textures throughout the interior basin, but nearly all of the microbial mats remain unlithified. The thin, muddy layer with poorly preserved, pigmented, laminated organic material—likely a decaying, buried mat—did not capture residual textures that would be identifiable in a microbialite (Fig. 2e). Cored sediments

were compositionally similar to the ooid-skeletal sand to gravel found on the open shoal and shoreface environments around Little Ambergris Cay, but the presence of minor amounts of mud in some intervals (Fig. 2e, Supplementary Fig. 2, and Supplementary Fig. 3) indicated that sedimentary textures developed in an episodically lower-energy environment. Angular crust intraclasts, cerithid gastropods, and minor mangrove debris—features unique to the interior basin—were found throughout the cores (Supplementary Fig. 4). This indicates that ooids present in vibracores formed in a shoreface or shoal environment and were

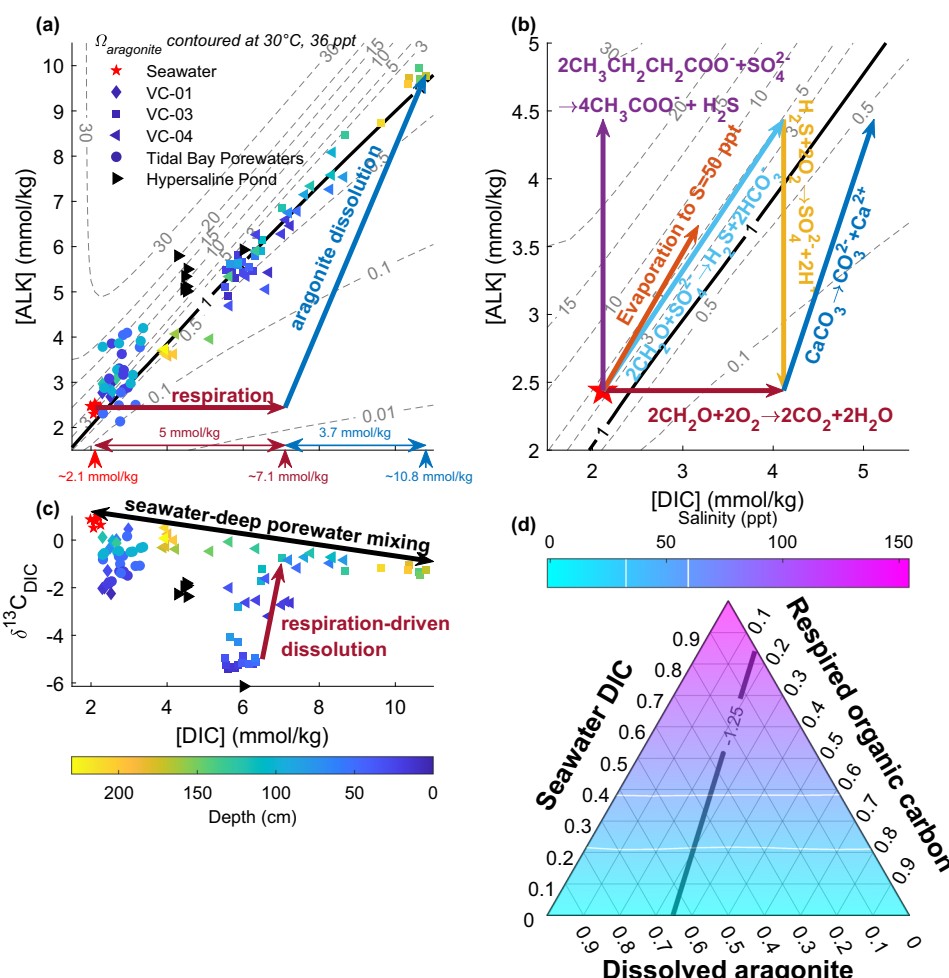

**Fig. 5 Processes impacting porewater carbonate mineral saturation.** Symbol shapes indicate sample location, symbol colors show depth, and arrows mark the effect of biogeochemical processes on DIC and alkalinity (ALK). **a** DIC and alkalinity data show that dissolution of ooid-skeletal sand contributed to increasing porewater carbon content and maintained aragonite saturation. If respiration-driven dissolution occurred by completely oxidizing organic matter, then the highest DIC, deep porewater must have ~5 mmol/kg of respired carbon. **b** Geochemical reactions that may non-uniquely sum to produce the DIC and alkalinity relationship observed. Aerobic respiration (red) and sulfide oxidation to sulfate (yellow) lower $\Omega$, while aragonite dissolution (dark blue), anaerobic respiration by sulfate reduction (light blue), incomplete anaerobic respiration, and fermentation of organic matter to dissolved organic matter by sulfate reduction (purple), and evaporation (orange) increase $\Omega$. **c** DIC concentration and DIC $\delta^{13}C$ data show that core tops (shallower than ca. 100 cm) are dominated by DIC from respired organic material and core bottoms contain increasing amounts of DIC from dissolved aragonite sand. **d** Ternary diagram of inorganic carbon sources in the highest DIC, deep porewater (10.8 mmol/kg), which has a $\delta^{13}C$ value of $-1.25‰$. It cannot have more than ~30% contribution from seawater DIC without exceeding observed salinities (white breaks in the color scale), so isotope mass balance indicates that it cannot have more than about 30% (~3 mmol/kg) contribution from respired carbon. This discrepancy with the stoichiometry of respiration-driven dissolution in panel **a** implicates a leak of unoxidized organic matter such as from the production of dissolved organic matter by incomplete anaerobic respiration and fermentation.

transported to and reworked in a restricted environment by storms, tidal currents, and bioturbation. The modern interior basin substrate is dominated by mats, so we infer that mats were present in the past but that these were not preserved beyond the rare horizons of decaying organic material. We cannot resolve when mats first colonized Little Ambergris Cay, nor predict if a future sedimentary package overlying the currently accumulating massive ooid-skeletal grainstone might be more diagnostic of mat colonization. Nonetheless, lithification of the extant mats is inhibited and therefore their preservation is unlikely.

Two synergistic processes limit lithification in the shallowest subsurface: efficient organic matter decay and tidal pumping of oxidants, dissolved organic matter, and sulfide. Aerobic respiration and oxidation of sulfide to sulfate decrease $\Omega$ by adding DIC, removing alkalinity, or both. In response to this, aragonite sand dissolves, adding DIC and alkalinity to subsurface waters until porewaters reach aragonite saturation (Fig. 5a). Tidal pumping

enables net aragonite dissolution (Fig. 4). At low tide, sulfate reduction occurs beneath a thermally stratified upper porewater lid, and saline, sulfidic porewater with high $\Omega$ seeps from the shallow subsurface into tidal channels. At high tide, seawater seeps into the shallow subsurface, oxidizing sulfide and driving aragonite undersaturation. Net dissolution of carbonate sediments occurs despite supersaturated platform waters because $O_2$ is imported to the sulfidic, shallow subsurface by tidal pumping.

In core porewaters, sulfate depletions relative to chlorinity represent net microbial sulfate reduction in excess of the observed sulfide concentrations. In iron-poor carbonate sediments, most of the sulfide produced is reoxidized[24,25]. Sulfate excesses of up to 18% indicate that sulfide is not always reoxidized in situ, as tidal advection moves sulfidic water through the subsurface and imports oxic and sulfate-rich seawater into the subsurface. Mangrove root xylem tissue on Little Ambergris Cay contains sulfate with $^{34}S/^{32}S$ nearly 30‰ lower than that of seawater

sulfate, suggesting that the oxygenic mangrove roots also facilitate sulfide oxidation in the porewater[26]. Sulfidic groundwater dynamics on Little Ambergris Cay, therefore, differ from those in sulfidic tidal ponds in fine-grained, muddy, sedimentary environments, in which the delivery of oxidants are limited by diffusion and consequently the fraction of sulfide that is reoxidized is low[27].

Carbon isotope measurements provide a means to identify the carbon sources in porewater. Inorganic carbon sources in this system include seawater DIC that has a $\delta^{13}C$ value of ~0.85‰, ooid sand that has a $\delta^{13}C$ value of ~5‰, and atmospheric $CO_2$ that has a $\delta^{13}C$ value of ~−8‰[16,28]. Organic matter in the microbial mats has a $\delta^{13}C$ value of ~−13‰[20,26] (Supplementary Table 2). In the shallowest porewater, which had the lowest DIC $\delta^{13}C$ values (Fig. 5c), carbon isotope mass balance dictates that about half of the DIC derived from reoxidized organic material and the remainder from subequal parts seawater and dissolved carbonate, consistent with the stoichiometry of respiration-driven dissolution driven by complete oxidation of organic matter to $CO_2$ (Fig. 5b). Slightly lower $\delta^{13}C$ values of carbonate sand within the uppermost 50 cm of cored sediments (Fig. 3) additionally suggest that a low $\delta^{13}C$ phase may precipitate in the shallowest sediments and/or preferentially dissolve at depth; this may be the selective dissolution of micrite envelopes and solution enlargement of microbores (Fig. 2d and Supplementary Fig. 4)[29].

Porewater compositions evolved with depth from the shallow endmember (comprised of DIC from respired organic matter and respiration-driven dissolution of aragonite) towards a mixture of seawater and a high DIC endmember (10.8 mmol/kg) that has a $\delta^{13}C$ value of −1.25‰ (Fig. 5c). Porewater salinities were normal marine to hypersaline (Fig. 3), indicating that atmospheric $CO_2$ dissolved in fresh, meteoric water did not drive significant dissolution or contribute measurably to the subsurface DIC inventory. Similarly, seawater-derived DIC would have had to have been concentrated nearly three-fold by evaporation for its admixture to satisfy the porewater $\delta^{13}C$ composition without additional organic carbon or dissolved carbonate input (Fig. 5d), but no such high-salinity waters were observed (Fig. 3). Thus, further dissolution of carbonate sand in the subsurface below the mats contributed to the DIC with high $\delta^{13}C$ values.

The stoichiometry of respiration-driven dissolution makes the prediction that the deep porewater in equilibrium with aragonite should have nearly 50% of the DIC from respired carbon (Fig. 5a), but the DIC $\delta^{13}C$ budget indicates that only approximately 30% of DIC derived from respired organic matter (Fig. 5d). To reconcile this, carbon must leak out of the subsurface such that respiration drives dissolution but does not contribute much to the porewater DIC inventory. For example, incomplete oxidation of butyrate or lactate by sulfate-reducing microorganisms produces acetate and sulfide (Fig. 5b). These soluble organic species could flush out during ebb tides but would not oxidize as readily as the sulfide during tidal inundation with oxic waters—a process that occurs rapidly even in the absence of microbial catalysis. This leak of partially oxidized, dissolved organic material out of the Little Ambergris system was observed as plumes of brown, tannic porewater seeping from tidal channels (Supplementary Fig. 6).

These results suggest a taphonomic bias against forming microbialites from microbial mats in environments like Little Ambergris Cay. Across the vast majority of Little Ambergris Cay, expansive mats developed throughout the island interior do not lithify despite widespread inorganic precipitation of ooid sand, crusts, and hardgrounds. The mat ecosystem hosts taxa that both build the mats and selectively degrade their organic material and ecologic structures[20]. In addition, over the entirety of the island's surface and subsurface ecosystem, microbial processes result in net dissolution that inhibits cementation of any mat-derived textures—and, in fact, dissolves both autochthonous and allochthonous carbonate—regardless of any microbial processes within the mats that may modulate calcium carbonate precipitation.

To drive lithification, processes such as evaporation or photosynthesis must create enough carbonate mineral supersaturation that cement precipitation rates exceed the rate saturation is decreased by aerobic respiration and sulfide oxidation to sulfate[7,30]. At a steady-state, microbial mats host communities that must produce approximately the same amount of biomass as they decay[4]. On Little Ambergris Cay, oxidation and decay on tidal, diurnal timescales, therefore, implies that these mats must have high gross productivity to sustain the rapid rates of respiration. Little Ambergris Cay is dominated by processes inhibiting lithification—facilitated by tidal advection—that result from this microbial ecosystem's high gross productivity. In a system open to atmospheric $CO_2$ and atmosphere-derived and seawater-derived oxidants ($O_2$ and sulfate), steady-state microbial ecosystems performing processes that might have aided carbonate precipitation (e.g., oxygenic photosynthesis and anaerobic respiration) continuously create products (e.g., organic matter and sulfide) that attenuate the saturation-increasing processes and instead enable net dissolution. Microbial metabolisms contributing to this process occur both within and below the mats. This results in interior basin surface waters that have lower aragonite saturation states (4.3) than the surrounding platform waters (4.5–5.6), in spite of higher salinities and extensive benthic photosynthesis[16]. Consequently, the thick, luxuriant mats present across Little Ambergris Cay are not preserved in the subsurface.

Although net productive ecosystems may lithify in some environments[9,10], high gross primary productivity in a steady-state, open-system environment provides a high flux of products that may enable net dissolution. For example, highly gross and net-productive mangrove forests produce large amounts of peat but dissolve carbonate sediments[31]. Perhaps unsurprisingly, ancient carbonate rocks rarely contain evidence for textures that look like lithified microbial peats[32]. From the perspective of preservation, a robust microbial ecosystem can be too much of a good thing—fueling rapid respiration and preventing early lithification.

Given a certain carbonate mineral saturation state as a boundary condition, factors that would promote microbialite preservation include higher organic carbon burial efficiency, less $O_2$ delivery, or more iron (permitting sulfide removal as pyrite). The limited encrusted domes along the pond rim (Fig. 2f), interpreted as lithified mats, provide an important example. Recalcitrant plant material in the pond may decay more slowly than the microbial mats, or saline, turbid conditions in the pond may limit robust mat growth and respiration compared to the interior basin. Incomplete oxidation of organic matter and diminished tidal advection permit calcification to proceed. Similarly, sediment-rich mats, such as those forming subtidal stromatolites in other Bahamian tidal channels, may also have less productivity and slower respiration, allowing ambient conditions to control lithification[6,10].

Seafloor cementation and sediment trapping and binding have demonstrable roles in microbialite formation. While microbial mats are the sine qua non of microbialites, our observations suggest that mat productivity also influences preservation. Mats in sandy, high-energy depositional environments with permeable sediments and advective water fluxes that supply plentiful $O_2$ may be especially prone to decay. Our data indicate that oxidation of microbial products can overcome an extrinsic, supersaturated seawater boundary condition and prevent lithification of widespread, robust microbial mats. Somewhat paradoxically, perhaps

ancient mats that had the highest gross productivity are rare or absent in the rock record.

## Methods

**Sediment and water sampling**. Vibracores were obtained by clamping 10 ft. core barrels (3″ OD, 0.083″ wall thickness aluminum 6063 irrigation tubing) to a concrete vibrator (Oztec H250OZ 2.5″ × 13″ steel pencil head) driven by a portable gasoline motor (Honda GX-160, 5.5 hp) with an 18 ft. flexible shaft (Oztec FS18OZ). The vibrating barrels were sunk into the sediment by a team of handlers using guide ropes tied to the top of the barrel and sealed with a 3″ gripper plug prior to recovery. No core catchers were used, and core recovery in the grainy, non-cohesive sediments was maximized by rapidly pulling the cores from the ground without mechanical assist. Core compaction and recovery were calculated from measurements of core barrel penetration, depth of core top prior to gripper plug emplacement, and length of recovered sediment (Supplementary Table 1). Recovered cores were laid flat, excess barrel length removed with a reciprocating saw, and sealed with low-density polyethylene pipe covers (Caplugs SC-3 sleeve caps) and electrical tape.

At each tidal bay porewater sampling station, 3/8″ polyvinyl chloride tubing (1/4″ ID, McMaster-Carr Masterkleer 5233K56) with 60 µm polyester filter fabric (McMaster-Carr Ultra-thin Filter Fabric 92255T72) covering the buried end was set to a depth of 20 cm, 50 cm, and 100 cm. Surface water and water from each tube at each depth were sampled at both low and high tide with a hand-operated vacuum pump. Tubing was also suspended over a hypersaline pond at the southern end of the cay at 0, 20, 40, 60, 80, 100, and 140 cm depths to sample the water column geochemical structure. Porewaters from vibracores were pulled into syringes from porous ceramic water samplers (Rhizosphere Rhizon CSS 19.21.23 F) inserted into holes drilled through the core casing at 3 to 10 cm intervals.

Porewaters and pond waters were aliquoted through 0.2 µm syringe filters into vials for geochemical analyses. Approximately 10 mL were dispensed into plastic tubes for temperature, pH, and salinity measurement and discarded. Temperature and pH were measured by an electrode (WTW 3310 pH meter with SenTix 41 pH probe) calibrated with NBS buffers daily at pH 4, 7, and 10. Salinity was estimated with an optical refractometer (ATAGO S-28E). Two mL were dispensed into microcentrifuge tubes with 100 µL of 3% zinc acetate for sulfate and sulfide concentration measurement, 4 mL were dispensed into borosilicate glass vials with conical polypropylene-lined caps for titration of total alkalinity, and 1 mL was dispensed via a needle into helium-flushed 12 mL borosilicate vials (Labco Exetainer 9RK8W) with 100 µL of 42% $H_3PO_4$ and sealed with chlorobutyl septa (Labco VW101) for determination of DIC concentration and $\delta^{13}C$ values. Following pore fluid sampling, cores were split with a circular saw on a jig with the blade depth set to the tube wall thickness, followed by a wire to split the sediment into even halves. Graphic logs of vibracore sedimentology, prepared with Sedlog[33], are shown in Supplementary Fig. 2 and Supplementary Fig. 3.

**Aqueous chemistry measurements**. DIC concentrations and $\delta^{13}C$ values were determined using a Thermo Fisher Scientific Gasbench II and Thermo Scientific Delta V Plus continuous flow mass spectrometer[34] with an 88% He dilution of headspace $CO_2$. Concentrations were determined from a bicarbonate solution calibration curve, and carbon isotopic compositions determined relative to Vienna Pee Dee Belemnite (V-PDB) with three in-house calcite reference materials used for detector linearity and drift correction. Precision and accuracy of DIC concentrations were determined with Dickson seawater reference material, and of $\delta^{13}C$ with a fourth in-house carbonate reference material. DIC concentration precision was between 1% and 5% (1 relative standard deviation [s.d.]) and $\delta^{13}C$ precision between 0.03‰ and 0.1‰ (1 s.d.).

Total alkalinity was determined by Gran titration at Caltech using a Metrohm 907 Titrando[35,36]. Samples were diluted to a total volume of 16 mL and then titrated using 0.01 mL injections of 0.025 M HCl. Measurement accuracy was monitored using both Dickson seawater reference material and an in-house standard composed of 19.148 meq/kg $NaHCO_3$ solution in 0.5 N NaCl. Replicate measurements of a 1.310 meq/kg total alkalinity artificial seawater standard were used to assess acid and electrode drift. Reproducibility was 0.15% (1 relative s.d.), corresponding to less than 28 µeq/kg for field samples and the high-alkalinity in-house standard and 2 µeq/kg for Dickson seawater.

Aragonite saturation states were calculated from DIC and alkalinity measurements using a MATLAB implementation of CO2SYS[37,38]. Measured temperatures and salinities were used when available or approximated from nearby data as described in the Supplementary Data. Aragonite saturation indices, $\Omega$, were calculated from the temperature and salinity-dependent stoichiometric solubility product, $K_{sp}^*$, by assuming that calcium is conservative with salinity; $\Omega = 1$ at aragonite saturation:

$$\Omega = \frac{[Ca^{2+}][CO_3^{2-}]}{K_{sp}^*} \tag{1}$$

This treatment neglected non-conservative deviations in the calcium ion concentration, such as due to carbonate precipitation or dissolution.

Sulfate and chloride concentrations were determined on an ion chromatograph (Dionex DX-120) at Johns Hopkins. Standards were prepared by dilution of Multi

Ion anion IC standard solution in $H_2O$ (Dionex Specpure 35565). Precision was 0.5% (1 relative s.d.) for sulfate and 0.7% (1 relative s.d.) for chloride-based on replicate measurements. To examine the effect of sulfate reduction or sulfide oxidation on sulfate concentration changes apart from those caused by evaporation, sulfate anomalies are reported as the percent deviation from normal seawater as:

$$100 \times \left[ \left( \frac{[SO_4^{2-}]}{[Cl^-]} \right)_{measured} \Big/ \left( \frac{[SO_4^{2-}]}{[Cl^-]} \right)_{seawater} - 1 \right] \tag{2}$$

Sulfide concentrations were determined spectrophotometrically via the Cline method[39] at Johns Hopkins. Sulfide standards were made on the day of analysis by dissolving sodium sulfide nonahydrate and zinc chloride in water deoxygenated by purging with $N_2$ gas and stored on ice. Samples were diluted with deoxygenated water and stored on ice. Sulfide reagents (LaMotte 3654-01-SC Sulfide Test) were mixed immediately prior to addition. Absorbances were measured at 670 nm on a Thermo Scientific Genesys 10 S UV-Vis Spectrophotometer. Precision was 5% (1 relative s.d.) based on replicate measurements.

**Carbonate mineral stable isotope measurements**. For carbonate mineral $\delta^{13}C$ and $\delta^{18}O$ analyses, carbonate grains from 2 to 4 g of sediment were sieved to <250 µm, rinsed with deionized water, and dried overnight in a 50–80 °C oven. Ooids were picked from bulk sediment under a dissecting microscope. Samples of both sieved, washed bulk sediment and the ooid fraction were powdered and 250–350 µg were placed into 12 mL round-bottom borosilicate vials (Labco Exetainer 938 W), which were subsequently flushed with helium. Carbon dioxide was generated by reaction with 100 µL >100% $H_3PO_4$ at 25 °C in a Thermo Fisher Scientific Gas-Bench II heating block at 30 °C overnight at Johns Hopkins. $\delta^{13}C$ and $\delta^{18}O$ compositions were determined on a Thermo Scientific MAT253 gas isotope ratio mass spectrometer in continuous flow mode relative to in-house standards calibrated to NBS-18 and IAEA-603 and are reported relative to V-PDB. Reproducibility was better than 0.1‰ for $\delta^{13}C$ values and 0.2‰ for $\delta^{18}O$ values based on replicate measurements of standards.

## Data availability

All data generated or analyzed during this study are included in this published article and its Supplementary Information files. Source data for Figs. 3, 4, and 5 are provided in the Supplementary Information files.

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

## Acknowledgements

We thank the Agouron Institute and Simons Collaboration on the Origins of Life for funding; the Turks and Caicos Islands Department of Environment and Coastal Resources for Scientific Research Permit 17-06-02-12; the Tarika family, P. Mahoney, J. Seymour, and J. Grancich for logistical support; A. Simms and L. Reynolds for coring advice; K. Dawson, G. Ames, F. Wu, A. Kupfer-Harris, and D. Brenner for lab and analytical support; and J. Alleon, A. Bahniuk, H. Grotzinger, K. Metcalfe, D. Morris, E. Orzechowski, D. Quinn, C. Sanders, E. Sibert, and J. Strauss for assisting fieldwork. E.J.T. and M.L.G. were further supported by NASA Exobiology grant 80NSSC18K0278.

## Author contributions

T.M.P., J.P.G., M.L.G., E.J.T., N.T.S., U.F.L., M.T.T., W.W.F., and A.H.K. conceived of the research, designed and executed the field campaign, collected cores and samples, and collected pH, salinity, and temperature data in the field. T.M.P., M.L.G., E.J.T., and J.N. collected geochemical data in the lab. E.J.T. prepared sediment thin sections and photomicrographs. T.M.P and M.L.G. processed the geochemical data. T.M.P., M.L.G., E.J.T., N.T.S., U.F.L., J.N., M.T.T., M.D.C., W.W.F., A.H.K., and J.P.G. interpreted the data. T.M.P. prepared the figures. T.M.P. wrote the manuscript with input from all authors.

## Competing interests

The authors declare no competing interests.
