## [Peer Review File · Nature Communications]

REVIEWER COMMENTS

Reviewer #1 (Remarks to the Author):

Review of NCOMMS-20-39593-T

Comments for the author

This manuscript seeks to explain the puzzling observation that microbialites are not preserved in Little Ambergris Cay, despite abundant microbial mat development and apparently favorable conditions for carbonate lithification. The principal conclusion of the manuscript is that aerobic heterotrophic processes, specifically aerobic respiration and sulfide oxidation, combined with tidal pumping of oxygen-rich waters, result in geochemical conditions conducive to carbonate *dissolution* and organic decomposition. This geochemical situation results in loss of organic-rich mat horizons prior to carbonate growth/cementation intervals.

The study provides compelling, though not unambiguous, evidence that tidal pumping and aerobic microbial processes combine to prevent microbialite formation. Although the relationship between aerobic respiration and carbonate dissolution is well-known, this field-based study is the first, to my knowledge, that combines porewater geochemistry, observations of Recent microbial mats and subsurface cores, and simple geochemical modeling to lend insight to the taphonomic bias against microbialite preservation in modern open marine environments. I recommend that this manuscript be published, although there are places that would benefit from clarification, addition of information, and strengthening of interpretations. I detail the most important below and additionally attach an annotated version of the manuscript with marginal comments that highlight places I found confusing or problematic.

In some places, the geochemical data is presented in a confusing way or is incomplete or misleading. The conclusions of this paper emerge from a pretty limited, though overall robust, dataset, and it's crucial to make sure that your reader understands the strengths and limitations of the data. This journal format is extremely space-limited, which means every word is precious and the figures are packed with information. Therefore, organization is key. The attached annotated manuscript highlights some places where changes could be made, but my recommendations fall into a few categories. First, group data types together. In discussing the vibracore data, for example (starting L97), discuss all the vibracore data at once – don't go back and forth among vibracore, groundwater, and pond data. Second, describe *trends* and apply meaning to them. For example, L71 makes a claim that the sediment $\delta^{13}\text{C}$ values range between 3.8 and 5.2‰. This isn't helpful, because that range isn't related to the stated depth trend in the next part of the sentence. Just describe the trends and be clear that one of the three cores does not exhibit the trend that is described. Third, explicate your geochemical conclusions just a little more. The brevity of the paper makes it difficult to commit to using extra words, but it's necessary here. The figures are extraordinarily data-rich and it's going to take your reader a long time to unpack them. Help us out by stating the boundary conditions up front and clearly. I did not find any flaws in the geochemical model, but I really had to work at it to figure out how you arrived at the particular boundary conditions you used. Either make use of the supplemental information or put the assumptions in the text/figure captions.

Be clear in the distinction between porewater and groundwater. At the beginning of the manuscript, it was clear to me that the term porewater referred to fluids collected from vibracore and that groundwater was obtained from piezometer wells on the SW side of the island. At some level, both kinds of samples are porewaters (as far as I can tell, you haven't got a water table here...) but the method of collection, as well as the relationship to the tidal cycle, is different. A brief explanation regarding the differences would also be helpful.

Some additional data needs to be added in order to have a complete picture. All the data described in the text must be included in the data table. In particular, sulfate anomaly data isn't present as a column in the data table, nor is the calculation of sulfate anomaly described anywhere in the manuscript. Data collected from the hypersaline pond is neither included in a figure nor in the data table. In addition, there appear to be a few values used from the literature and not measured directly in this study (e.g., isotopic composition of microbial mat organic matter; seawater composition). These are crucial to the geochemical models developed in this paper and their values (and citations) should all be collected together and included in a supplemental data table.

The overall explanation of tidal pumping + aerobic metabolisms is well supported by the porewater data and field observations. The details of the geochemical model are shaky, though, and are not as well supported by the data. In the end, the manuscript argues that high productivity by mats produces the conditions that lead to their lack of preservation. But high productivity (which isn't established directly) doesn't necessitate high decomposition rates. Of course, the presence of biomass is a necessary prerequisite for respiration and sulfate reduction to occur. The simplest interpretation of the data is that the continuous supply of oxygen to the site of the microbial mats allows for aerobic process to dominate the decomposition of these mats. If the mats were sealed off from O₂, anaerobic processes, which are less carbonate-dissolving, would come to dominate. This appears to be occurring, at least transiently, in the hypersaline pond. In short, this isn't a closed system – in fact, organic matter from mangroves is likely a factor as well. It's not unreasonable to ignore mangrove carbon in the geochemical modeling, but it is worth noting that any additional carbon added to this system and respired aerobically would additionally contribute to carbonate dissolution.

I like the framing of the preservation of a microbialite as essentially a taphonomic process. Build on this idea a little more, by clarifying what kind of preservation you're talking about and when each set of taphonomic processes is in play. There are two taphonomic processes acting on the mats: (1) preservation of organic matter and organic mat structure and (2) mineralization of that structure to create a microbialite. The first is essentially biological taphonomy – if mat decomposition occurs fast, the time window to mineralize a microbialite is short, and if mat structure is lost, no amount of carbonate precipitation will get it back. The second is taphonomy based on mineral precipitation – if mineralization happens before the mat is decomposed, no microbialite. For favorable overall microbialite taphonomy, organic decomposition must be slow(ish) and mineralization must be fast(ish), relative to one another. What this manuscript concludes is that Ambergris Cay mats represent the opposite – fast(ish) organic decomposition and slow(ish) [actually, negative] mineralization. I think it might help to frame these two sets of processes separately. They're linked (in fact, the fast decomposition causes the slow lithification), and that's a key conclusion of this paper, but it's handy to

think of them separately, particularly if one would like to speculate about alternative environments that *could* produce a microbialite.

It's worth noting that the local presence of a microbial mat isn't necessary to produce carbonate-dissolving behavior. The groundwater analyses also show low aragonite saturation in places. See the comment on L211. Even without microbial mats here, aragonite dissolution might still be occurring.

Some comments keyed to the text (more comments in the manuscript copy):

L17 – I recommend revising the final sentence of the summary, as primary productivity itself isn't proposed as the mechanism by which microbialite development is inhibited. Indeed, primary productivity alone, has the opposite effect, as pointed out by the authors. See comment.

L50 – Later in the paper, you make the statement that microbial mat productivity is “high”. It might be helpful to estimate what proportion of these environments are microbial mat-covered. It seems that the assertion of high productivity is based on the widespread nature of these mats, so it would be helpful to describe that a little more explicitly here.

Figure 1 – The drone image in panel b gives a misleading perspective on topography. In particular, VC-03 looks like it's high above the waterline. I had to go to Google Earth to realize the actual shape of the cay. I'm not sure how this figure could be modified to show the level of inundation, but I encourage the authors to consider some strategies.

Figure 3 – It would be helpful to label panels a-c as low supratidal; intertidal; and subtidal. The groundwater depiction is substantially different from the porewater depiction, leaving me wondering why these are included in the same figure. This may be a journal constraint on number of figures; if it's not, I recommend separating panel d into another figure. The white line in the 2 top-right panels of (d) is not explained; nor are the white dots in the panel depicting aragonite saturation at low tide. I'm also confused about the datum for the groundwater stations. How can the cross section be exposed in some of the panels and not in others, even at the same tidal stage? This needs a little more explanation.

Figure 4 is a lot to take in. See marginal comments on the manuscript copy for additional detailed comments. Panel (c) needs a little more work, I think. First, the seawater-groundwater mixing line, as drawn, is confusing to me. It's not a mixing line between the groundwater measured in this study, because the groundwater measured in the study sits within the red circle I drew. So, mixing between that groundwater and seawater won't produce the line shown.

Instead, the mixing line appears to connect seawater to deep pore water. However, in the text, mixing along this line isn't how the observed [DIC] and $\delta^{13}\text{C}$ values are produced. There's a suggestion in the manuscript that the red respiration-driven dissolution arrow highlights a pathway for evolution of porewaters; however, that pathway isn't the pattern shown by the data. Both VC-03 and VC-04 data look more complex than that (paths shown in yellow highlighter) VC-01 data is more difficult to interpret, but might follow a seawater-groundwater mixing pattern or a respiration-driven dissolution

path. This is key, I think. There's a limited amount of data here and it supports your hypothesis generally, but there's a lot of noise in the dataset – naturally. The explanation of these data feels like an over-interpretation of a relatively noisy dataset.

L197 – I see what you're getting at here, but I don't see how that solves your "carbon gap". If 50% of the DIC present at depth had to come from respired carbon (in order to get the amount of DIC you observe), leaking organic carbon out of the system won't solve that mass balance problem. I'm not sure how you solve it, theoretically, though. It might be enough to point out that there's a gap here. It doesn't undermine your main conclusions, but it is puzzling.

L206 – here, sulfate reduction is called out as a key process in lack of microbialite preservation, but this hasn't been previously developed.

L211 – I don't think the case has been made for high gross productivity. There's no data for rates of microbial mat growth or photosynthetic activity. This really isn't central to the paper's conclusions, though; I'd recommend softening the language here.

L301 – I'm not sure that treating Ca^{2+} conservatively in the geochemical modeling is fair. Given the amount of CaCO_3 dissolution invoked, Ca won't be behaving as a conservative element. At the very least, this should be addressed in the text, as non-conservative behavior could confound the straightforward explanations posed here.

In summary, this is an important paper, and the conclusions are supported by the evidence. The overall conclusion that tidal pumping delivers oxygen to fuel aerobic, carbonate-dissolving processes is sound, though the details could use some refinement, particularly in terms of the limits of geochemical modeling given this dataset. This manuscript has the potential to frame thinking about the genesis of microbialites throughout Earth history and additionally provides a testable framework within which to interrogate modern microbialites. If the authors can address the issues I raise above, the article should be published. –Julie Bartley

Reviewer #2 (Remarks to the Author):

Sometimes what isn't there is more interesting than what is. That is certainly the case with the report by Present et al, which present a story of microbialites that are not what is expected. Where environmental conditions are favorably, lithified microbialites are expected. However, in the Little Ambergris Cay, the authors have found that microbialites, despite seemingly favorable environmental conditions, do not lithify. They go on to present a thorough description of the geochemical and geological conditions at the site that suggest a unique confluence of biogeochemical events inhibit lithification. Given the importance of interpreting both present and paleo signatures of biogenic and abiotic lithification for understanding the evolution of the earth system, further investigation of the microbialite ecosystem of Little Ambergris Cay may yield exciting findings. I think that this paper will serve to both bolster and inspire those investigations.

The main complaint I have of this paper is its heavy use of jargon. Given the potential broad readership of Nature Communications, I urge the authors to consider if their use of jargon is always necessary.

Reviewer #3 (Remarks to the Author):

This paper makes a strong and well-argued case that there has been a sizable taphonomic bias in the geological record of most productive microbial mats and of the associated sedimentary fabric in the form of microbialites. This of course echoes some of Robert Riding's earlier conclusions on anaerobic metabolisms favouring microbial mat lithification. In my view, the one (minor) weakness of the paper is that it assumes the dominance of microbial mats in the interior basin in the past based on the presence of a thin muddy layer with poorly preserved organic material in the core. I would expect to see a biolaminated fabric in this sediment. The authors mention the evidence for storms, tidal currents, and bioturbation (line 135) as some of the agents affecting sedimentation in the interior basin. Could these agents simply inhibit the formation of microbial mats? Or destroy the biolamination? Other than that, this paper meets all the criteria for validity, significance and originality. A possibility that mat productivity influences its preservation in the fossil record is of strong interdisciplinary interest, whereas any new information elucidating the temporal patterns in carbonate precipitation is always a major breakthrough in the field of geobiology. I have no objections, and warmly recommend publication of this paper. Dmitriy Grazhdankin

Reviewer #1:

This manuscript seeks to explain the puzzling observation that microbialites are not preserved in Little Ambergis Cay, despite abundant microbial mat development and apparently favorable conditions for carbonate lithification. The principal conclusion of the manuscript is that aerobic heterotrophic processes, specifically aerobic respiration and sulfide oxidation, combined with tidal pumping of oxygen rich waters, result in geochemical conditions conducive to carbonate *dissolution* and organic decomposition. This geochemical situation results in loss of organic-rich mat horizons prior to carbonate growth/cementation intervals.

The study provides compelling, though not unambiguous, evidence that tidal pumping and aerobic microbial processes combine to prevent microbialite formation. Although the relationship between aerobic respiration and carbonate dissolution is well-known, this field-based study is the first, to my knowledge, that combines porewater geochemistry, observations of Recent microbial mats and subsurface cores, and simple geochemical modeling to lend insight to the taphonomic bias against microbialite preservation in modern open marine environments. I recommend that this manuscript be published, although there are places that would benefit from clarification, addition of information, and strengthening of interpretations. I detail the most important below and additionally attach an annotated version of the manuscript with marginal comments that highlight places I found confusing or problematic.

In some places, the geochemical data is presented in a confusing way or is incomplete or misleading. The conclusions of this paper emerge from a pretty limited, though overall robust, dataset, and it's crucial to make sure that your reader understands the strengths and limitations of the data. This journal format is extremely space-limited, which means every word is precious and the figures are packed with information. Therefore, organization is key. The attached annotated manuscript highlights some places where changes could be made, but my recommendations fall into a few categories. First, group data types together. In discussing the vibracore data, for example (starting L97), discuss all the vibracore data at once – don't go back and forth among vibracore, groundwater, and pond data. Second, describe *trends* and apply meaning to them. For example, L71 makes a claim that the sediment $\delta^{13}\text{C}$ values range between 3.8 and 5.2‰. This isn't helpful, because that range isn't related to the stated depth trend in the next part of the sentence. Just describe the trends and be clear that one of the three cores does not exhibit the trend that is described. Third, explicate your geochemical conclusions just a little more. The brevity of the paper makes it difficult to commit to using extra words, but it's necessary here. The figures are extraordinarily data-rich and it's going to take your reader a long time to unpack them. Help us out by stating the boundary conditions up front and clearly. I did not find any flaws in the geochemical model, but I really had to work at it to figure out how you arrived at the particular boundary conditions you used. Either make use of the supplemental information or put the assumptions in the text/figure captions.

We thank the reviewer for the careful, thoughtful, and constructive comments. We have adopted the suggestion to reorganize our presentation of results and highlight trends in each category of data (entirely rewritten porewater results section at Lines 106-130). We also more clearly explain the carbon isotopic compositions of endmember carbon sources, which are summarized in Lines 198-199, with a new table in the Supplemental Information (Supplemental Table 2).

Be clear in the distinction between porewater and groundwater. At the beginning of the manuscript, it was clear to me that the term porewater referred to fluids collected from vibracore and that groundwater was obtained from piezometer wells on the SW side of the island. At some level, both kinds of samples are porewaters (as far as I can tell, you haven't got a water table here...) but the method of collection, as well as the relationship to the tidal cycle, is different. A brief explanation regarding the differences would also be helpful.

Thank you for highlighting this difference. Throughout, we previously tried to refer to pore fluids from the vibracores as "porewater" and pore fluids collected *in situ* from the piezometers in the tidal bay as

“groundwater.” This distinction was because we were indeed looking for evidence of a water table with the piezometers, which we did not find, but still treated that category of data as groundwater collected with both temporal and spatial resolution. This was confusing because we also used “groundwater” to talk about the deepest porewaters in the vibracores that appeared to be an endmember geochemical composition. Upon revision, we ceased to use “groundwater.” Throughout the manuscript and figures, we now refer to the data collected from the piezometers as “tidal bay porewaters” or “shallow subsurface porewaters,” and to the deep, high-DIC porewater endmember as “deep porewater.”

Some additional data needs to be added in order to have a complete picture. All the data described in the text must be included in the data table. In particular, sulfate anomaly data isn't present as a column in the data table, nor is the calculation of sulfate anomaly described anywhere in the manuscript. Data collected from the hypersaline pond is neither included in a figure nor in the data table. In addition, there appear to be a few values used from the literature and not measured directly in this study (e.g., isotopic composition of microbial mat organic matter; seawater composition). These are crucial to the geochemical models developed in this paper and their values (and citations) should all be collected together and included in a supplemental data table.

Data from the hypersaline pond was already tabulated in the submitted data table under the heading, “Pond Limnology.” To clarify, we have added a plot of the data to the Supplemental Information (Supplemental Figure 6). We have also added a column of the sulfate anomaly, as well as clarified how we calculated it from sulfate and salinity data in the Methods section (Line 340-342). Finally, the $\delta^{13}\text{C}$ data we compiled from the literature were all previously listed, with citations, at Lines 198-199. As described above, we additionally placed these into a table, which we have added to the Supplementary Information as Supplemental Table 2.

The overall explanation of tidal pumping + aerobic metabolisms is well supported by the porewater data and field observations. The details of the geochemical model are shaky, though, and are not as well supported by the data. In the end, the manuscript argues that high productivity by mats produces the conditions that lead to their lack of preservation. But high productivity (which isn't established directly) doesn't necessitate high decomposition rates. Of course, the presence of biomass is a necessary prerequisite for respiration and sulfate reduction to occur. The simplest interpretation of the data is that the continuous supply of oxygen to the site of the microbial mats allows for aerobic process to dominate the decomposition of these mats. If the mats were sealed off from O_2 , anaerobic processes, which are less carbonate-dissolving, would come to dominate. This appears to be occurring, at least transiently, in the hypersaline pond. In short, this isn't a closed system – in fact, organic matter from mangroves is likely a factor as well. It's not unreasonable to ignore mangrove carbon in the geochemical modeling, but it is worth noting that any additional carbon added to this system and respired aerobically would additionally contribute to carbonate dissolution.

We thank the reviewer for highlighting these sources of confusion because they are critically important interpretations of our data. First, we agree that we do not directly document high mat productivity (although, as described below, we do infer that they are fast-growing and productive primary producers). However, we do directly observe—surprisingly—that the mats decay at least as quickly as they grow, and the organic matter decomposition occurs on tidal (diurnal) timescales such that there are always products of microbial metabolisms (i.e. organic matter and sulfide) that can be oxidized, driving large and temporally-variably aragonite saturation gradients. We note that our sulfate, sulfide, and pore fluid data indicate that continuous supply of oxygen is not available to the mats or to the shallow subsurface. Instead, we observe tidal advection of seawater (with oxygen and sulfate) and groundwater (with sulfide and dissolved organic matter) into and out of the shallow subsurface—both direct aerobic respiration of organic matter and sulfuric acid production by sulfide oxidation (which is enabled by sulfide production by anaerobic respiration of organic matter) drive carbonate dissolution. To see net carbonate dissolution, as we do on Little Ambergris Cay, indeed requires an open system for both CO_2 and O_2 and/or SO_4^{2-} : the carbon fixed by the mats cannot just come from seawater DIC and the O_2 cannot just be locally produced by photosynthesis, or the reaction stoichiometry of respiration-driven dissolution (directly by

aerobic respiration or indirectly via sulfide oxidation of anaerobic respiration by-products) simply reverses the effect of photosynthesis. Instead, respiration-driven dissolution of pre-existing carbonate sand is driven by addition of CO₂ from the atmosphere and O₂ and/or SO₄²⁻ from tidal advection of seawater into the system. We worked to clarify the importance of these processes in the Discussion at Lines 163-168 and 235-244. Finally, we agree that organic carbon derived from mangroves and respired in the manner just described would similarly drive carbonate dissolution. Mangroves appear to be a minor component of the ecosystem on Little Ambergris Cay compared to the microbial mats, as measurements of bulk sediment organic matter δ¹³C from across Little Ambergris Cay in the immediate vicinity of mangrove trees (Raven et al., 2019) are indistinguishable from microbial mat organic matter measured by Gomes et al. (2020), which we now summarize in Supplemental Table 2, suggesting that microbial mat organic matter dominates that from mangroves.

I like the framing of the preservation of a microbialite as essentially a taphonomic process. Build on this idea a little more, by clarifying what kind of preservation you're talking about and when each set of taphonomic processes is in play. There are two taphonomic processes acting on the mats: (1) preservation of organic matter and organic mat structure and (2) mineralization of that structure to create a microbialite. The first is essentially biological taphonomy – if mat decomposition occurs fast, the time window to mineralize a microbialite is short, and if mat structure is lost, no amount of carbonate precipitation will get it back. The second is taphonomy based on mineral precipitation – if mineralization happens before the mat is decomposed, no microbialite. For favorable overall microbialite taphonomy, organic decomposition must be slow(ish) and mineralization must be fast(ish), relative to one another. What this manuscript concludes is that Ambergris Cay mats represent the opposite – fast(ish) organic decomposition and slow(ish) [actually, negative] mineralization. I think it might help to frame these two sets of processes separately. They're linked (in fact, the fast decomposition causes the slow lithification), and that's a key conclusion of this paper, but it's handy to think of them separately, particularly if one would like to speculate about alternative environments that *could* produce a microbialite.

This is a very helpful framework, and we adopted the suggestions at Line 225-232. Gomes et al. (2020) studied the preservation of organic matter within and immediately below the mats. At Lines 144-146, we also tried to rephrase the essential paradox of Little Ambergris Cay and many other microbial mat environments: a steady-state microbial mat biomass empirically exists in a supersaturated environment that is precipitating abundant minerals such as calcium carbonate, but the mats are not preserved. Here, we show how the mats and their bioproducts, counterintuitively, enable a measurable and significant decrease in the aragonite supersaturation that apparently is enough to slow mineralization to less than that of decay rates. We agree with the reviewer's important observation that decomposition rates and mineralization rates are directly linked. Our key results is that the decomposition doesn't just undo the favorable biogeochemical processes that promote lithification—in the open system studied here, the microbes in fact decrease the aragonite saturation state of the environment, accounting for the strong bias against microbialite preservation.

It's worth noting that the local presence of a microbial mat isn't necessary to produce carbonate dissolving behavior. The groundwater analyses also show low aragonite saturation in places. See the comment on L211. Even without microbial mats here, aragonite dissolution might still be occurring. Some comments keyed to the text (more comments in the manuscript copy):

We completely agree, and hope to clarify this important point—indeed, as summarized above, the metabolic products of the microbes in the entire surface-mat-groundwater system (especially sulfide) interact with tidal pumping of the island to decrease the saturation state of the depositional environment, inhibiting mat preservation despite platform conditions conducive to lithification. We added an explicit note of this at Line 243-244.

L17 – I recommend revising the final sentence of the summary, as primary productivity itself isn't proposed as the mechanism by which microbialite development is inhibited. Indeed, primary productivity alone, has the opposite effect, as pointed out by the authors. See comment.

Thank you for noting this confusion. We modified Line 18 to clarify that the primary productivity and respiration, together, interact to lower the environmental carbonate saturation state by replacing “primary productivity and respiration” with “microbial metabolisms.”

L50 – Later in the paper, you make the statement that microbial mat productivity is “high”. It might be helpful to estimate what proportion of these environments are microbial mat-covered. It seems that the assertion of high productivity is based on the widespread nature of these mats, so it would be helpful to describe that a little more explicitly here.

Thanks for this suggestion—Stein (2020) and Stein et al. (in review) mapped the facies distribution of Little Ambergris Cay with differential GPS and drone imagery and determined that the mat facies cover over 25% of the island. Barring bedrock, the mats cover more than 40% of peritidal environments. We have added these data to Line 41-43. In addition, repeat visits to Little Ambergris Cay by Stein et al. (in review) and Stein (2020) document substantial new mat growth and colonization of scoured ooid sand in the weeks and months following Hurricanes Irma and Maria in 2018, consistent with rapid productivity as the system returns to steady state.

Figure 1 – The drone image in panel b gives a misleading perspective on topography. In particular, VC-03 looks like it's high above the waterline. I had to go to Google Earth to realize the actual shape of the cay. I'm not sure how this figure could be modified to show the level of inundation, but I encourage the authors to consider some strategies.

We appreciate the surprising color variability and areal extent of the peritidal microbial mats on Little Ambergris Cay. To help readers discern truly supratidal land and shoal from the interior basins and tidal channels that we cored, we added a white contour of the maximum tidal level derived from a drone photogrammetric digital elevation model. This contour is added to the modified Figure 1 caption at Line 54-56.

Figure 3 – It would be helpful to label panels a-c as low supratidal; intertidal; and subtidal. The groundwater depiction is substantially different from the porewater depiction, leaving me wondering why these are included in the same figure. This may be a journal constraint on number of figures; if it's not, I recommend separating panel d into another figure. The white line in the 2 top-right panels of (d) is not explained; nor are the white dots in the panel depicting aragonite saturation at low tide. I'm also confused about the datum for the groundwater stations. How can the cross section be exposed in some of the panels and not in others, even at the same tidal stage? This needs a little more explanation.

We thank the reviewer for the helpful suggestions. We have labeled panels a-c by their tidal setting and substrate. We also split the contoured groundwater geochemical data into its own figure, now Figure 4. The white dots are small contours of $\Omega=1$, to which we have now added an annotation on the figure. Finally, the blank white space at high tide is because we, unfortunately, did not collect samples of surface seawater for DIC, alkalinity, chlorinity, or sulfate concentrations. We amended the figure to clarify which gaps are due to exposure (the two interior-most stations on the tidal bar at low tide), and which are due to lack of data.

Figure 4 is a lot to take in. See marginal comments on the manuscript copy for additional detailed comments. Panel (c) needs a little more work, I think. First, the seawater-groundwater mixing line, as drawn, is confusing to me. It's not a mixing line between the groundwater measured in this study, because the groundwater measured in the study sits within the red circle I drew. So, mixing between that groundwater and seawater won't produce the line shown. Instead, the mixing line appears to connect seawater to deep pore water. However, in the text, mixing along this line isn't how the observed [DIC] and $\delta^{13}\text{C}$ values are produced. There's a suggestion in the

manuscript that the red respiration-driven dissolution arrow highlights a pathway for evolution of porewaters; however, that pathway isn't the pattern shown by the data. Both VC-03 and VC-04 data look more complex than that (paths shown in yellow highlighter) VC-01 data is more difficult to interpret, but might follow a seawater-groundwater mixing pattern or a respiration-driven dissolution path. This is key, I think. There's a limited amount of data here and it supports your hypothesis generally, but there's a lot of noise in the dataset – naturally. The explanation of these data feels like an over-interpretation of a relatively noisy dataset.

As described above, we clarified our usage of porewater and ceased to refer to any waters as groundwater. We revised Figure 5c to show the fluid mixing between seawater and the deepest, high DIC porewater endmember. Indeed, the respiration-driven dissolution in the shallow parts of the cores (indicated by the red arrow) does not explain the composition of this deep, high DIC endmember. We explore the origin of this endmember in Figure 5d. To this end, we recognized that it's helpful to emphasize the break the cores at ca. 1 m depth, which we do with further modification of the caption. In upper parts of cores, DIC is set by respiration-driven dissolution. Deeper than 1 m, the deep porewaters are a mixture of seawater and dissolved aragonite sand, but the $\delta^{13}\text{C}$ budget indicates more dissolved aragonite than explainable solely from aerobic respiration-driven dissolution. In a narrow interval, these two zones mix in each core.

L197 - I see what you're getting at here, but I don't see how that solves your "carbon gap". If 50% of the DIC present at depth had to come from respired carbon (in order to get the amount of DIC you observe), leaking organic carbon out of the system won't solve that mass balance problem. I'm not sure how you solve it, theoretically, though. It might be enough to point out that there's a gap here. It doesn't undermine your main conclusions, but it is puzzling.

We thank the reviewer for understanding that the isotope mass balance and DIC-Alkalinity balance are difficult to reconcile but not central to our key conclusion that oxidation of microbial products (organic matter and sulfide) overcomes the extrinsic, supersaturated seawater boundary condition and prevents microbialite lithification. However, a DIC leak—a mechanism we suggest based on observations of high dissolved organic matter outflows (Supplemental Figure 5)—would indeed reconcile this carbon gap. The DIC $\delta^{13}\text{C}$ budget allows for no more than 30% of the DIC deriving from oxidized organic matter, but the stoichiometry of respiration-driven dissolution dictates that 50% must have come from oxidized organic carbon. The solution is to drive dissolution with respiration but not allow that respired carbon to enter the DIC pool; i.e., it partially oxidizes to dissolved organic (rather than inorganic) carbon visible as brown, tannic acids. We worked to clarify this at Lines 218-220.

L206 – here, sulfate reduction is called out as a key process in lack of microbialite preservation, but this hasn't been previously developed.

Sulfate is an important oxidant imported into the microbial ecosystem by tidal advection and is responsible for respiring much of the organic carbon and producing sulfide that can in turn be oxidized by O_2 to dissolve aragonite. We deleted this sentence and clarified this point in the paragraph discussing how open-system tidal advection of carbon and oxidants prevents mat lithification (Lines 239-243).

L211 – I don't think the case has been made for high gross productivity. There's no data for rates of microbial mat growth or photosynthetic activity. This really isn't central to the paper's conclusions, though; I'd recommend softening the language here.

In addition to the observed widespread nature and rapid regrowth of the mats as described above, we note that a key framework for understanding microbial mats is as steady-state ecosystems in which different communities produce and consume organic carbon and other biogeochemical products. For the mats to have a standing biomass, they must be producing at least as much as is consumed. We argue that the most productive microbial mats also produce the most byproducts (organic matter and sulfide) that inhibit lithification. We worked to clarify these implications at Lines 235-237.

L301 – I'm not sure that treating Ca^{2+} conservatively in the geochemical modeling is fair. Given the amount of CaCO_3 dissolution invoked, Ca won't be behaving as a conservative element. At the very least, this should be addressed in the text, as non-conservative behavior could confound the straightforward explanations posed here.

Carbonate dissolution would add Ca^{2+} to porewater. By not accounting for this, we calculate minimum estimates of aragonite saturation. Indeed, saturation states are close to equilibrium and slightly exceed it in parts of the subsurface, probably resulting in the nodular texture below ca. 1 m in the cores, but there is still net carbonate dissolution and lack of mat preservation on Little Ambergris Cay. Further, because Ca^{2+} is so much more abundant in seawater than DIC (~10 mmol/kg compared to ~2 mmol/kg), dissolution of ~9 mmol of CaCO_3 per kg porewater required to reach the highest observed DIC concentrations of 10.8 mmol/kg would only approximately double the Ca^{2+} concentration and therefore the calculated aragonite saturation from 1.2 to 2.4. This is still less than open platform saturations of 4.5 to 5.6 (Trower et al., 2018) despite evaporation and photosynthesis presumably driving tidal bay waters to higher saturation states.

In summary, this is an important paper, and the conclusions are supported by the evidence. The overall conclusion that tidal pumping delivers oxygen to fuel aerobic, carbonate-dissolving processes is sound, though the details could use some refinement, particularly in terms of the limits of geochemical modeling given this dataset. This manuscript has the potential to frame thinking about the genesis of microbialites throughout Earth history and additionally provides a testable framework within which to interrogate modern microbialites. If the authors can address the issues I raise above, the article should be published. –Julie Bartley

We greatly thank Dr. Bartley for the helpful comments!

Additional comments from the manuscript:

L44 – Are the crusts mineralized/cemented, or just dry organic material

We clarified that the crusts are cemented ooids at Line 43

L84 - Visible organic matter? It doesn't look like you measured TOC, so this is an observation, yes?

That is correct, and we added "visible" to Line 93.

L94 - Where were the decaying microbial mats observed in 2016? Within the calcareous dome structures/crusts?

Yes, large sheets of microbial mat lined the pond floor near its rim, where the dome structures are now. We clarified this relationship at Line 103.

L100 - As I look at the data table, I don't see a datum for each groundwater station – presumably the elevations of the ground surface in the bar vs channel are a bit different. I don't see any topography drawn in Fig. 3d – perhaps the change in elevation is too small to draw there? It would be good to clarify. Also, what's the tidal throw here and where is the water table within each station?

We adjusted Figure 3 as described above.

L101 - The same hypersaline pond that has the calcified domes?

Yes, we clarified this in Line 127-128.

L107 - Would be useful to summarize the trends – from supratidal to subtidal and with depth. Most variables don't show any particular trend, and that's worth noting. Maybe take just a little more space and reorganize this paragraph. Figure 3 is packed with information and this paragraph needs to help the reader unpack it.

L110 - I had a little trouble following the result reporting here. Because Figure 3 is organized by sample type (porewater, groundwater), not by analysis type, it is difficult to connect this paragraph with the data in Figure 3. I think it would be better, both for clarity and for the interpretations, to describe all the parameters with respect to porewater, then move on to groundwater, then finally pond water.

I think I'd recommend reorganizing both paragraphs in this section into three paragraphs (1) Porewater was samples.... Here are the results; (2) Groundwater was sampled...Here are the results (3) Hypersaline pond waters were sampled... Here are the results....

We implemented this framework to reorganize our discussion of the results and summarize trends in the data, as described in the response to the reviewer's comments above.

L112 - These data aren't in Figure 3; you should refer the reader to the data table, or add a panel.

We added reference to the data table and added a new supplemental figure as described above.

L127 - Most? The hypersaline pond seems to be the exception.

We qualified this as "but nearly all of the microbial mats remain unlithified" at Line 146.

L134 - This is maybe a little ambiguous (because of the dual meaning of 'cored'. Maybe 'This indicates that ooids present in vibracores formed in...'

We adopted this phrasing at Line 153.

L148 - It would be good to add the sulfate anomaly as a column in your data table.

We have added the sulfate anomaly column to our Supplemental Data table.

L148 - I don't understand this part of the sentence. What do you mean by in excess of the observed sulfide concentration?

In VC-04, I see a relationship between the sulfate anomaly and sulfide concentration. But I can't tell from the data how much sulfate is actually being oxidized here, and whether it's in excess of the change in sulfide observed. Also, the pattern you describe is really only discernable in one core.

It occurs to me, though, that given the degree of tidal pumping and the likelihood that sulfide oxidation and sulfate reduction aren't occurring simultaneously, I don't think the data help you discern a mass balance here.

We agree that sulfide oxidation is not occurring at the same time or place as sulfate reduction, as indicated by the lack of mass balance. We clarify at Line 170-172.

L150 - I think you need to define what the sulfate anomaly is and how it's calculated. I read this value as something entirely different – I assumed that seawater sulfate:chloride was set to an anomaly of 0 and that positive numbers were ppm sulfate above or below the anomaly. I see now that the units are %, but I'm having trouble visualizing percent of what base value. The calculation needs to be included.

We described how we calculated sulfate anomaly more thoroughly and what its units are at Line 340-342. An anomaly of 0 indeed means that the $\text{SO}_4^{2-}/\text{Cl}^-$ ratio is the same as seawater, and the units are relative change, in percent, from the $\text{SO}_4^{2-}/\text{Cl}^-$ ratio of seawater.

L173 - And organic matter from mangrove or other vascular plant debris is likely about-30%. You might also consider macroscopic algae as a source of organic matter, which also might be more negative than the (likely) C-limited microbial mats.

I don't think the argument posed here is necessarily wrong, but a little mangrove carbon could significantly alter mass balance calculations. I'd recommend calling it out as a source of uncertainty here.

We agree, although we note that addition of lower $\delta^{13}\text{C}$ organic matter would necessitate even more carbonate dissolution in excess of the stoichiometry of respiration-driven dissolution. That said, as described above, Raven

et al. (2019) measure bulk sediment organic matter $\delta^{13}\text{C}$ from directly below mangroves and find the $\delta^{13}\text{C}$ composition indistinguishable from the microbial mats reported by Gomes et al. (2020), which we document more explicitly in the new Supplemental Table 2. Mangrove-derived organic matter appears to be subordinate to mat-derived organic matter, both spatially and by mass, on Little Ambergris Cay.

L152 - You've switched to talking about groundwater here, so I'm not sure how this relates to the porewater data.

L183 - I don't understand the "seawater-groundwater mixing line" in 4c. The seawater and groundwater datapoints both lie toward the left side of this diagram. The yellow squares at high DIC represent the deepest porewaters.

L186 - ? porewater?

L190 - Porewater?

Figure 4d - Porewater?

As described above, we stopped using "groundwater" in various ways and explicitly clarified that the ternary diagram constrains the range of possible endmember mixtures that may contribute to the deepest, highest DIC porewater observed. We have clarified our usage of "porewater" and replaced usages of "groundwater" throughout the manuscript.

L203 - Here, you need to be clear about what kind of preservation you're talking about. There are two taphonomic processes acting on your mats (1) preservation of organic matter and organic mat structure and (2) mineralization of that structure to create a microbialite. (1) is essentially a biological equation – if mat decomposition occurs fast, the time window to mineralize is short. (2) is a mineral precipitation equation – if mineralization happens before the mat is decomposed, no microbialite. For favorable microbialite taphonomy, organic decomposition must be slow(ish) and mineralization must be fast(ish), relative to one another. What you're saying is that Ambergris Cay mats represent the opposite – fast(ish) organic decomposition and slow(ish) mineralization [actually dissolution, but that counts!]. I think it might help to frame these two sets of processes separately. They're linked (in fact, the fast decomposition causes the slow lithification), but it's handy to think of them separately, particularly if one would like to speculate about alternative environments that could produce a microbialite.

This is a very helpful framework that we have adopted as described above.

L206 - I'm a little confused here – sulfate reduction would not appear to be the culprit in driving aragonite saturation lower. Sulfate reduction likely doesn't dissolve aragonite, though it does dispose of organic matter quite nicely! Perhaps that's what you're getting at here and it just needs a little clarification.

At Lines 239-243, we worked to emphasize that sulfate reduction also produces sulfide, which may be oxidized by dioxygen in seawater and strongly reduce aragonite saturation. Figure 5b shows that the combined processes of sulfate reduction and sulfide oxidation result in the same net effect on aragonite saturation as aerobic respiration-driven dissolution, or incomplete organic matter oxidation followed by sulfide oxidation may cause a net decrease in aragonite saturation.

L210 - Yes. Based on the groundwater data, tidal advection is key. Aragonite saturation indices are low even in the groundwaters, away from the microbial mat growth regions. In fact, your lowest aragonite saturation values (0.1) occur in the tidal bay groundwaters at high tide. This argues against intrinsic microbial mat processes as driving the undersaturation – the organic matter to sustain high rates of aerobic respiration could come from anywhere – mangrove bits, macroscopic algae... or microbial mats. If there's a source of organic matter in an environment with high subsurface porosity and tidal pumping, your data would suggest that dissolution is inevitable.

That doesn't mean your conclusions are wrong – I think you've nailed it, but the microbial mats aren't the ultimate cause of this geochemical situation. Abundant oxygen availability is, delivered by tidal pumping and enhanced by the high porosity subsurface environment produced by development on an ooid platform.

We agree, although we emphasize that both key microbial metabolic products—organic matter and sulfide—enable this situation (Lines 242).

The hypersaline pond is the exception that proves the rule. Where hypersalinity-induced stratification limits tidal advection, less oxygen is delivered to the environment via pumping and aragonite saturation creeps up, ultimately producing crusts. This microenvironment, I think, gives insight to how microbialites might be produced despite unfavorable metabolic circumstances.

We agree that the lithified mats in the pond are the local exception to the rule. We reemphasize their minor but important significance at Lines 252-255.

L211 - I don't think you've made the case for high gross productivity, though I'd agree with the conclusion of low net productivity (or efficient recycling, regardless of gross productivity).

As described above, we interpret the mats as steady-state ecosystems in which the most productive microbial mats also produce the most byproducts (organic matter and sulfide) that would inhibit lithification. We worked to clarify this implication of high productivity at Lines 235-236.

L214 - I'm not sure where this comes from. It might be worth adding a data table that gives surface water values, even if they're from the literature. If you measured these in your study, I don't see the values in your data table.

The aragonite saturation states of platform seawater and one analysis from Little Ambergris Cay are published in Table 4 of Trower et al. (2018) and summarized at Line 244-245.

L221 - O₂ [or just oxygen]

We replaced "dioxygen" with "O₂" throughout, as suggested.

L301 - Is this a reasonable assumption, given the rather large contribution of carbonate dissolution to your porewaters? At the very least, this needs to be addressed in the text. In particular, it seems that both aerobic respiration and carbonate dissolution have the potential to release Ca²⁺ into porewaters, potentially confounding the straightforward explanations posed in this paper. I also worry somewhat about pH influences on this system, which also don't seem to be accounted for. This effect is likely less problematic, but it would be wise to address it.

To the reviewer's first point, we discuss the relatively minor effect of nonconservative Ca²⁺ compared to DIC and alkalinity changes on porewater chemistry above. To the second point, DIC and Alkalinity are two measurements that permit calculation of pH using the CO₂ system equilibria. By titrating to measure alkalinity, we measure total alkalinity (not carbonate alkalinity). However, unlike pH, DIC behaves conservatively in mixtures of different fluids, permitting simpler mass balance calculations like those applied here.

Figure 1b - There may not be too much you can do here, but I definitely get the wrong impression about topography from the drone footage – I had to go to Google Earth to see where the tidal channels are and understand where the seawater is on this island. VC-03 and VC-05 appear well above the high tide line when I look at this image; I can see that this isn't the case on a Google Earth image. Perhaps you can recolor areas below the high tide line as more blue than brown? Or create polygons that show supratidal, intertidal, subtidal environments?

As discussed above, we added a contour of the maximum high tide level to Figure 1b to help differentiate between intertidal and supratidal environments.

Figure 3 - When I look at Figure 3, I do not see this increase with depth in the uppermost 50 cm of the core. I see no data for the uppermost 20 or 30 cm. Then, 2 of the 3 cores show a $\delta^{13}\text{C}$ increase by 0.5 to 1.2‰; one core shows a negligible increase. The way this is written, it sounds like all of the cores have $\delta^{13}\text{C}$ values of ~3.8 at the top and increase to 5.2‰ by 50 cm. I recommend you rephrase this and refer reader to the figure.

We added a reference to Fig. 3 as suggested. The $\delta^{13}\text{C}$ of sediments in the cores are all 5.1 to 5.2‰ at depth, and rise from 3.8‰ to 4.9‰ in the first ca. 50 cm. We rephrased Line 79-80 to describe the trends more systematically.

Might be useful to label each graph (a-c) as Low supratidal; Intertidal; Subtidal

We revised Figure 3a-c to indicate the core settings as suggested.

Part d might be best represented as a separate figure, since it represents groundwater rather than porewater from cores.

We adopted the suggestion to break this panel into its own, new, Figure 4.

What are the white dots in the lower panel and the white outline of the two fields on the rightmost two top panels?

The white dots are small contours of $\Omega=1$, to which we have now added an annotation on the figure.

Figure 4 - Add a 1 to the dark line in panel (a) [like you did in panel (b)]

We moved the '1' on the bold line for aragonite equilibrium so that it won't be hidden by the data.

Looking at the data table, the highest-DIC groundwater has DIC=3.37. I don't understand this portion of the diagram. Ah, as I read down, I think I see. Highest DIC porewater is 10.8. Elsewhere in the paper, you distinguish between porewater and groundwater; maintaining that distinction might be useful here.

As described above, we clarified our usage of "porewater" and the deep, high-DIC endmember throughout.

Take this piece of explanation and put it in the general explanation for figure 4 (before part (a) is described). Include a little more explanatory text – symbol shape indicates sample location; arrows indicate calculated (theoretical) direction of change due to various processes.

We revised the Figure 5b caption to introduce the symbol shape, color, and arrows before each panel as suggested.

Figure 4b - This panel needs a legend with words – aerobic respiration; sulfate reduction... I like that you've included the reactions as well, as it makes it clear why DIC and TA change the ways they do, but the processes are referred to by name in the text. You can either make a color-coded legend, or simply give the information in the caption.

We revised the Figure 5b caption to guide the viewer through the color-coded processes and their affect on aragonite saturation at Lines 183-187.

Figure 4c - This panel needs additional explanation – what is the mixing line? It isn't mixing between your measured groundwater and open seawater, because all that data clusters at low DIC/high $\delta^{13}\text{C}$. It seems like it's a mixing line between seawater and deep porewaters in the vibracore; but what meaning does that have? You're not arguing that your porewaters are generated by mixing of these two end-members.

The red arrow also needs some explanation.

We have clarified this figure and our usage of "groundwater" throughout the manuscript. In Figure 5c, we hope to indicate that all porewaters $\delta^{13}\text{C}$ values are explained by three carbon isotope endmembers: respired organic matter, seawater, and the deepest porewaters in each core. This last endmember is different in each core. The

red arrow for respiration-driven dissolution reflects aerobic respiration and accompanying carbonate dissolution, seen clearly in the uppermost portions of cores. The lower portion of cores, below about 1 m depth, itself represents a mixture between platform seawater and dissolved carbonate sand unaccompanied by low $\delta^{13}\text{C}$ respired organic matter.

Reviewer #2:

Sometimes what isn't there is more interesting than what is. That is certainly the case with the report by Present et al, which present a story of microbialites that are not what is expected. Where environmental conditions are favorably, lithified microbialites are expected. However, in the Little Ambergris Cay, the authors have found that microbialites, despite seemingly favorable environmental conditions, do not lithify. They go on to present a thorough description of the geochemical and geological conditions at the site that suggest a unique confluence of biogeochemical events inhibit lithification. Given the importance of interpreting both present and paleo signatures of biogenic and abiotic lithification for understanding the evolution of the earth system, further investigation of the microbialite ecosystem of Little Ambergris Cay may yield exciting findings. I think that this paper will serve to both bolster and inspire those investigations.

The main complaint I have of this paper is its heavy use of jargon. Given the potential broad readership of Nature Communications, I urge the authors to consider if their use of jargon is always necessary.

We thank the reviewer for the encouraging comments. This is an important suggestion. We tried to increase readability but maintain precision with the following minor edits:

- Line 35: replaced "allochems" with "grains"
- Line 155-158: rewrote sentence to avoid using "facies"
- Line 174: replaced "euxinic" with "sulfidic"
- Line 201: replaced "remineralized" with "reoxidized"
- Line 251, 261: replaced "dioxygen" with " O_2 "

Reviewer #3:

This paper makes a strong and well-argued case that there has been a sizable taphonomic bias in the geological record of most productive microbial mats and of the associated sedimentary fabric in the form of microbialites. This of course echoes some of Robert Riding's earlier conclusions on anaerobic metabolisms favouring microbial mat lithification. In my view, the one (minor) weakness of the paper is that it assumes the dominance of microbial mats in the interior basin in the past based on the presence of a thin muddy layer with poorly preserved organic material in the core. I would expect to see a biolaminated fabric in this sediment. The authors mention the evidence for storms, tidal currents, and bioturbation (line 135) as some of the agents affecting sedimentation in the interior basin. Could these agents simply inhibit the formation of microbial mats? Or destroy the biolamination? Other than that, this paper meets all the criteria for validity, significance and originality. A possibility that mat productivity influences its preservation in the fossil record is of strong interdisciplinary interest, whereas any new information elucidating the temporal patterns in carbonate precipitation is always a major breakthrough in the field of geobiology. I have no objections, and warmly recommend publication of this paper. Dmitriy Grazhdankin

We thank Dr. Grazhdankin for these supportive comments. We note that our key finding is a surprising lack of lithification, despite favorable anaerobic metabolisms and environmental conditions. We added a reference to Dr. Riding's work at Line 26. Dr. Grazhdankin also notes that we do not know if the mats dominated the depositional environment in the past, which we indicate at Lines 154-158. First, indeed, crinkly laminated sedimentary structures are often associated with microbialites and interpreted as biolamination and we observe—in a single muddy layer—possible evidence of such a fabric. Notably, throughout the cores, the sediments were mud-poor and grainy; laminations of any type are often difficult to discern, although we show

minor evidence of cross lamination in Figure 2c. Second, while high-energy events such as storms certainly destroy many mats, they do not seem to inhibit mat (re)growth. For example, Stein (2020) and Stein et al. (in review) document rapid regrowth of microbial mats in the months following the direct passage of Hurricane Irma's Category 5 eyewall over Little Ambergris Cay. Nonetheless, our results are not predicated on preservation of past mats—our primary finding is rapid decay and a lack of lithification of the microbial mats currently growing across the island despite the abundant inorganic precipitation of aragonite on the Caicos platform and the local, surficial enhancement of saturation states by evaporation and photosynthesis.

References cited for review:

- Gomes, Maya L., Leigh Anne Reidman, Shane S. O'Reilly, Usha F. Lingappa, Kyle S. Metcalfe, David A. Fike, John P. Grotzinger, Woodward W. Fischer, and Andrew H. Knoll. "Taphonomy of Biosignatures in Microbial Mats on Little Ambergris Cay, Turks and Caicos Islands." *Frontiers in Earth Science* 8, no. 576712 (2020): 1–22. <https://doi.org/10.3389/feart.2020.576712>.
- Raven, M. R., D. A. Fike, M. L. Gomes, and S. M. Webb. "Chemical and Isotopic Evidence for Organic Matter Sulfurization in Redox Gradients Around Mangrove Roots." *Frontiers in Earth Science* 7 (2019). <https://doi.org/10.3389/feart.2019.00098>.
- Stein, Nathaniel Thomas. "Investigation of Past Habitable Environments through Remote Sensing of Planetary Surfaces." Ph.D., California Institute of Technology, 2020. <https://resolver.caltech.edu/CaltechTHESIS:06092020-122543624>.
- Stein, Nathan T., John P. Grotzinger, Daven P. Quinn, Usha F. Lingappa, Theodore M. Present, Elizabeth J. Trower, Maya L. Gomes, et al. "Geomorphic and Environmental Controls on Microbial Mat Fabrics on Little Ambergris Cay, Turks and Caicos Islands." *Sedimentology*, in review.
- Trower, Elizabeth J., Marjorie D. Cantine, Maya L. Gomes, John P. Grotzinger, Andrew H. Knoll, Michael P. Lamb, Usha Lingappa, et al. "Active Ooid Growth Driven By Sediment Transport in a High-Energy Shoal, Little Ambergris Cay, Turks and Caicos Islands." *Journal of Sedimentary Research* 88, no. 9 (September 30, 2018): 1132–51. <https://doi.org/10.2110/jsr.2018.59>.

REVIEWER COMMENTS

Reviewer #1 (Remarks to the Author):

I recommend publication of this manuscript. The revisions as submitted satisfactorily address all of my major concerns. I appreciate the adjustments in terminology and organization made by the authors - the relationships between data and conclusions are easier to follow now. I think this paper makes an extremely important contribution to our understanding of microbialite genesis (or lack thereof!). --
Julie Bartley

Reviewer #3 (Remarks to the Author):

The authors have revised their manuscript in the light of comments received by prior external review; the interpretations and conclusions are now both succinct and clearer. I would like to see this manuscript published, but I am also intrigued with the way the manuscript ends. How do the authors define 'the most productive ancient mats' (line 263)? Based on the amount of biomass produced and stored? Indeed, microbial communities exhibiting the highest net rates of primary productivity should produce the most biomass. The net primary productivity is calculated as gross primary productivity minus the rate of energy loss to metabolism and ecosystem maintenance. The Little Ambergris Cay microbial mats having low net productivity, therefore, by definition are not very productive. In addition, could the authors specify a modern sequence architecture of the Little Ambergris Cay depositional system? Is it a prograding peritidal system? Have the authors considered a possibility that their conclusions could be biased by the preservation of time-averaged microbial communities? I suggest the authors briefly address these issues in the manuscript. Other than that, I recommend this interesting and important manuscript for publication. Dmitriy Grazhdankin

Reviewer #1:

I recommend publication of this manuscript. The revisions as submitted satisfactorily address all of my major concerns. I appreciate the adjustments in terminology and organization made by the authors - the relationships between data and conclusions are easier to follow now. I think this paper makes an extremely important contribution to our understanding of microbialite genesis (or lack thereof!). --Julie Bartley

We sincerely thank Dr. Bartley for the helpful previous review and these encouraging comments!

Reviewer #3:

The authors have revised their manuscript in the light of comments received by prior external review; the interpretations and conclusions are now both succinct and clearer. I would like to see this manuscript published, but I am also intrigued with the way the manuscript ends. How do the authors define 'the most productive ancient mats' (line 263)? Based on the amount of biomass produced and stored? Indeed, microbial communities exhibiting the highest net rates of primary productivity should produce the most biomass. The net primary productivity is calculated as gross primary productivity minus the rate of energy loss to metabolism and ecosystem maintenance. The Little Ambergris Cay microbial mats having low net productivity, therefore, by definition are not very productive. In addition, could the authors specify a modern sequence architecture of the Little Ambergris Cay depositional system? Is it a prograding peritidal system? Have the authors considered a possibility that their conclusions could be biased by the preservation of time-averaged microbial communities? I suggest the authors briefly address these issues in the manuscript. Other than that, I recommend this interesting and important manuscript for publication. Dmitriy Grazhdankin

We thank Dr. Grazhdankin for his support and interest, and for raising a very important point to clarify. We mean that mats with the most gross primary productivity may be most susceptible to taphonomic bias by their own ecosystem dynamics; we modified line 267 to clarify that we are referring to gross productive mats. We believe a key insight from our observations on Little Ambergris Cay is that microbial mats—as part of a steady-state ecosystem whose constituents consume each other's products—produce the most sulfide and organic carbon when they have high gross productivity. This, in turn, permits open-system (e.g., tidal pumping) advection of those products and of environmental oxidants to result in a net diminishment of aragonite saturation, despite environmental conditions otherwise conducive to mineralization. Previous work that we cite in the introduction (ref. 9-11) has shown that net productive ecosystems are more prone to preservation. The key new insight from our

work is that ecosystems with high gross productivity, perhaps regardless of their net productivity, can create a significant taphonomic bias against mineralization. Indeed, some non-microbial ecosystems such as mangroves are hugely net and gross productive in aragonite-supersaturated environments, but similar dynamics result in dissolution of carbonate and accumulation (but not lithification) of peats. We added a reiteration of this framework and the example of mangrove peats in Australia to lines 248-250.

A complete model of Little Ambergris Cay's stratigraphic architecture is beyond the scope of this work, but a fuller description and interpretation of facies arrangements and temporal evolution is forthcoming in Stein et al., (*in review*). In brief, Little Ambergris Cay is a Holocene island accreting at the windward end of a 20 km-long ooid shoal forming in the lee of Big Ambergris Cay (Fig. 1). Peritidal and supratidal facies are indeed prograding as the island grows via accretion of high energy shoreface grainy sediments, beach ridges, and eolian dunes. Radiocarbon analyses of *Strombus* shells throughout the grainstone berm presented in Stein (2020) suggest that the entire island is Holocene in age, consistent with Dr. Grazhdankin's suggestion of progradational deposition of peritidal sediments. On the interior side of the island's eolian grainstone rim, sediment transport is dominated by storm deposition and scouring, and by tidal current reworking: it is currently unclear if the interior basin is growing in area and/or filling with sediment. Our cores include angular ooid grainstone (crust) intraclasts and high-spired gastropod shells—evidence for proximal restricted environments—at least 1.7 m below the current surface; while these may have been transported from the interior basin to the core sites while they were still shoal or shoreface depositional environments, these indicate that a restricted environment at least existed nearby.

Our subsurface geochemical data independently demonstrate that the modern microbial ecosystem results in aragonite dissolution and diminishment of preservation potential despite widespread aragonite precipitation on the Caicos platform—regardless of past microbial mat extent and facies arrangements, of their evolution through time as Little Ambergris Cay developed, and of the validity of assuming that microbial mats must have been previously present as evidence for their lack of preservation. We clarified that our results demonstrate this bias against preservation of the modern mats independently of the absence of preserved ancient mats at Lines 265-267.

References for review:

- Stein, Nathaniel Thomas. "Investigation of Past Habitable Environments through Remote Sensing of Planetary Surfaces." Ph.D., California Institute of Technology, 2020.
<https://resolver.caltech.edu/CaltechTHESIS:06092020-122543624>.
- Stein, Nathan T., John P. Grotzinger, Daven P. Quinn, Usha F. Lingappa, Theodore M. Present, Elizabeth J. Trower, Maya L. Gomes, et al. "Geomorphic and Environmental Controls on Microbial Mat Fabrics on Little Ambergris Cay, Turks and Caicos Islands." *Sedimentology*, in review.